# Flu Vaccination among Healthcare Professionals in Times of COVID-19: Knowledge, Attitudes, and Behavior

**DOI:** 10.3390/vaccines10081341

**Published:** 2022-08-18

**Authors:** Corrado Colaprico, Eleonora Ricci, Andrea Bongiovanni, Valentin Imeshtari, Vanessa India Barletta, Maria Vittoria Manai, David Shaholli, Mattia Marte, Pasquale Serruto, Giuseppe La Torre

**Affiliations:** Department of Public Health and Infectious Diseases, Sapienza University of Rome, 00185 Rome, Italy

**Keywords:** flu vaccination, healthcare professionals, COVID-19, knowledge, attitudes, behavior

## Abstract

The seasonal flu vaccine is the most important way to prevent influenza epidemics, so it is useful to increase the awareness of health professionals. The aim of our study is to evaluate knowledge, attitudes, and behavior about flu vaccination among healthcare professionals in times of COVID-19. Methods: A cross sectional study was carried out between November 2020 and April 2021. Participants were recruited in teaching hospital ‘Policlinico Umberto I’ of Rome. A survey of 24 questions about flu vaccination was administered, evaluating their knowledge, attitude, and practice about this topic. Results: 872 healthcare professionals were involved in the study (36.9% men, 63.1% women). More than 90% of the HCWs, especially physicians, recognize the importance of getting influenza vaccination: The main reasons for not getting vaccinated were fear of vaccine side effects (20.3%) and fear of the needle (6.4%). Nevertheless, 40.7% of the healthcare providers consider mandatory vaccination as unethical, especially if they work in low-intensity wards. Conclusion: a high percentage of healthcare workers agree with the importance of influenza vaccination and only a small percentage is still opposed. It is therefore important to continue to promote the influenza vaccination through communication and health education programs.

## 1. Introduction

Flu virus is a life-threatening respiratory virus, especially for people with a very fragile immune system. In general, it attacks the upper respiratory tract and in a few cases the bronchopulmonary section. The course can be characterized by fever, headache, asthenia, cough, and rhinitis [1], with symptoms lasting 1 or 2 weeks and no need for causal treatment or hospitalization. A different scenario applies for hospitalized patients, who can have more serious consequences than the average population. It depends on the age and on the immunocompromised state and it is worsened by pre-existing medical conditions, resulting in secondary bacterial pneumonia [2,3]. Published data show that every year influenza epidemics affect 5–15% of the world’s population, resulting in 4–5 million serious cases and 250.000 to 5000.000 deaths [4]. The most important prevention measure against seasonal influenza is vaccination and it is strongly recommended to healthcare workers (HCWs) by the WHO [5], the CDC [6], and Ministry of Health of Italy [7]. In Europe, the vaccination coverage is different, looking at three seasons 2015/2016, 2016/2017 and 2017/2018 it goes from 16% to 63% (median 30%) [8]. Belgium, England and Wales at the high position, Italy and Norway at the lowest [9].

For all these reasons, the Centers of Disease Control and Prevention highly recommends that all healthcare professionals must be vaccinated against influenza [10,11]. The rationale can be explained with the following three points.

The first is to reduce the risk of patients catching influenza from HCWs, the transmission of flu virus from HCWs is often the main source of nosocomial influenza outbreaks [12,13]. Although there are standard infection control precautions, such as hand hygiene and respiratory etiquette, which may reduce the spreading from infected HCWs to patients, they are not infallible or universally applied [14].

The second point is to protect HCWs and their families that have a high risk of exposure to virus. Kuster SP et al. carried out a meta-analysis that shows how among all influenza-infected persons not vaccinated, the influenza incidence rate was 18.7% among HCWS (RR 3.4) against other workers not associated with the health care setting [15].

The last one is to reduce the absenteeism during the winter season, causing personnel shortages in medical facilities, and as a result increasing costs for the national health service. As confirmed by an Italian study, the overall direct costs in Italy per self-reported influenza case is estimates as €56 [16], and influenza vaccination among HCWs, from a perspective of economic evaluation, is cost saving (€12 per vaccinee) in the case base scenario [17].

Every hospital has to consider a vaccination strategy based on these grounds valuing favorable cost-benefit and cost-effectiveness profiles [18].

Although there are similar studies of these subjects in the other countries, our study focuses on the value, knowledge, attitude, and behaviour of HCWs of a teaching hospital in Rome.

## 2. Materials and Methods

### 2.1. Study Design

A cross sectional study, according to the Strengthening the Reporting of Observational Studies in Epidemiology (STROBE) statement [19], was carried out between November 2020 and April 2021.

### 2.2. Setting and Sample

Participants were recruited in teaching hospital ‘Policlinico Umberto I’ of Rome. The following professionals were considered: doctors, nurses, radiology technicians, biologists, and other health professionals.

A survey of 24 questions about flu vaccination was administered online, inviting the potential participants (3471) via email and evaluating their knowledge, attitude, and practice about this topic. Each question has four answers, assessing how much the worker agrees with the statement. The questionnaire showed a good reliability (Cronbach alpha = 0.738).

Sample size calculation was carried out using EpiCalc2000 with the following parameters:oProportion of willingness to get vaccinated among HCWs: 47.00% (mean between values reported in other studies) [20,21]oNull hypothesis value: 52.00%oSignificance: 0.05oPower: 80%

The sample size needed was 769. In order to increase the power, a 10% increase was calculated. So, the final sample size calculation gave the need to recruit at least 846 HCWs.

### 2.3. Statistical Analysis

Statistical analysis was performed using mean and standard deviation (SD) for quantitative variables. For qualitative variables, frequencies and percentages were computed.

The four answers were dichotomized as a favorable or contrary opinion.

Differences between the answers were assessed using Chi-square test for univariate analysis.

Multivariate analysis was conducted using backward multiple logistic regression analysis considering the following explanatory variables: gender, age (< or > of 40 years), years of service, marital status (married or not), and role (doctor or other healthcare workers). The answer of each question was considered as a dependent variable. The choice of gender as possible explanatory variable is due to the highest likelihood of vaccine hesitancy among women [22], while age and job experience were chosen since increasing levels of these variables are associated to different odds of getting flu vaccination [23,24,25].

Results of the logistic regression models were presented as Odds Ratio with 95% confidence interval (95% CI).

All statistical analysis were performed using SPSS for Windows (IBM, Armonk, NY, USA). A statistically significant difference was accepted at a *p*-value of <5%.

## 3. Results

A total of 872 HCW were enrolled in our study and complied to the questionnaire administered among all the subjects, 322 were men (36.9%) and 550 (63.1%) women. The average age was 41.7 years (14.0 SD). Regarding the length of service, the median is 16.9 (13.3%). Regarding the civil status, 335 (38.4%) participants were divorced or separated, 287 (32.9%) were single, 192 (22%) were cohabitants, and 335 participants (38.4%) were married. Finally, 446 (51.1%) were doctors, 126 (14.4%) were health care workers, 232 (26.6%) were nurses, 14 (1.6%) auxiliaries, and 31 (3.6%) technicians (Table 1).

### 3.1. Univariate Analysis

The results of the univariate analysis (Table 2a–c) show that the question of whether healthcare professionals believe that vaccination against influenza is a professional responsibility for the operators themselves who can unintentionally transmit the flu, endangering patients’ lives, 93.9% of doctors and 82.9% of other healthcare workers answered significantly “Yes”. In total, 89.6% of operators said they would have undergone voluntary flu vaccination as a protection for themselves and the people they are in contact with. Most of the participants (88.8% with *p* < 0.001) would undergo mandatory vaccination if it were offered directly in the workplace, of these, 94.8% are doctors and 82.4% are other healthcare workers. Less than half of the interviewees (40.7% *p* < 0.001) believed that obliging operators to get vaccinated is unethical as it violates the individual’s choice; 26.6% believed they should not be obliged if they work in an operating unit with low risk, even if healthy (25.6%), and that it is not necessary to be vaccinated if the hygiene rules are respected (16.3%). A low percentage of operators was afraid of the injection (6.4%), of side effects (20.3%), and believed that the latter can be dangerous (23.9%). Of the 872 healthcare workers involved, 68.1% believed that the flu is dangerous only for the elderly and with previous illnesses and 75.8% said that vaccinated subjects are not at risk of transmitting flu to their patients.

Finally, almost all of the interviewees believed that operators should promote vaccination (90.7%) and that influenza vaccination is important to prevent the disease (94.4%). Table 2 describes the questions administered to the study group with their respective answers.

### 3.2. Multivariate Analysis

The results of the multivariate analysis (Table 3a–c) show that operators over the age of 40 believe that influenza is dangerous only for people over 65 (OR 1.69; 95% CI: 1.07–2.67), who are not at risk of transmitting flu, when vaccinated, to patients (OR 1.47; 95% CI: 1.07–2.03), and tend to be afraid of the side effects of the flu vaccine (OR 1.90; 95% CI: 1.33–2.71). They claim that health education programs organized by the working structure can be useful (OR 2.15; 95% CI: 1.14–4-08) and tend to have had the flu vaccination (OR 1.69; 95% CI: 1.27–2.25).

Married or cohabiting healthcare professionals tend not to be afraid of injection (OR 0.50; 95% CI: 0.29–0.87) and argue that vaccination should only be recommended for healthcare professionals (OR 1.54; 95% CI: 1.15–2.08), unlike doctors that believe that vaccination anti-influenza is a professional responsibility of healthcare professionals (OR 3.21; 95% CI: 2.01–5.13) and that it would be right if it was mandatory (OR 3.03; 95% CI: 2.18–4.22). Doctors tend to respond positively to the proposal to undergo mandatory vaccination if it is offered in the workplace (OR 3.61; 95% CI: 2.19–5.97) or voluntary to protect themselves and the people they come in contact with (OR 3.56; 95% CI: 2.17–5.85). In line with what has been said, doctors believe that anti-flu vaccination is important to prevent the disease (OR 3.00; 95% CI: 1.52–5.91) and that when vaccinated they do not risk transmitting the flu to patients (OR 2.73; 95% CI: 1.96–3.82). Doctors themselves tend to affirm that health professionals must promote vaccination because they are responsible for the health education of the citizen (OR 5.49; 95% CI: 3.06–9.84) and having had a flu shot (OR 2.72; 95% CI: 2.04–3.63).

Doctors, when asked whether obliging health workers to vaccinate is unethical since it violates the individual’s choice, tend to answer negatively (OR 0.35; 95% CI: 0.27–0.47); similarly, they do not feel that they should not be obliged to be vaccinated if they work in a low-risk operating unit (OR 0.43; 95% CI: 0.32–0.59), if they are in good health (OR 0.42 CI 0.31–0.58), or if the anti-contagion hygiene rules are respected (OR 0.32; 95% CI: 0.21–0.47).

Lastly, doctors are not afraid of the side effects of vaccination (OR 0.20; 95% CI:0.13–0.29) and tend not to believe that the flu vaccine can have dangerous side effects (OR 0.33; 95% CI: 0.24–0.46).

On the basis of length service in the structure to which they belong, operators tend to believe that influenza vaccination is not a professional responsibility of health professionals (OR 0.45; 95% CI: 0.29–0.70) and that they would generally not undergo mandatory vaccination if it were offered at the place of work (OR 0.44; 95% CI: 0.28–0.68). On the contrary, they believe that only healthcare workers over the age of 65 should be obliged (OR 1.78; 95% CI: 1.34–2.36) and that forcing healthcare professionals to get vaccinated would be unethical as it would violate the individual’s choice (OR 1.40; 95% CI: 1.06–1.86).

## 4. Discussion

Our study aimed at exploring the knowledge, attitudes, and behavior toward influenza vaccination in a sample of healthcare workers (HCWs) of an Italian teaching hospital in times of COVID-19.

Italy is an interesting country to consider on this issue, due to the low attitudes and adherence toward flu vaccination [26]. The flu coverage rate in the pre-pandemic period was between 4 and 13% [27], and this survey demonstrates the high degree of flu vaccine trust and willingness to get vaccinated, much more than those reported in other studies (between 40.8% and 53.8%) [21,22,28].

The CDC consider flu vaccination as the first and most important measure to protect against flu viruses. More than 90% of the HCWs in our study recognize the importance of getting and promoting influenza vaccination and almost all of them would undergo voluntary vaccination to protect themselves, their family, and surrounding community, in line with finding from other studies [29].

HCWs, especially physicians and those over 40 years old, are aware that >65-year aged people or with previous illness are at more high risk of influenza and its consequences and tend to have had flu vaccination. On the other hand, 16.3% believe that it is not necessary to vaccinate if anti-county hygiene rules are followed. It is well known that implementing hygiene measures helps from getting infected or spreading flu viruses, but it is not enough. Health professionals work at close contact with patients, which increases the risk for transmitting the infection. Moreover, long work hours and shift work increase the risk for reduced job performance and increase the possibility of fatigue-related errors [30]. Furthermore, Alhumaid et al., in their systematic review, identified several gaps in HCWs knowledge about occupational vaccination, among which is influenza [31].

The main reasons for not getting vaccinated were fear of vaccine side effects (20.3%) and fear of the needle (6.4%). Overall, almost one quarter of the participants believe that the flu vaccine can have dangerous side effects. This belief tended to be more present among health professionals older than 40 years old. On the contrary, physicians and those who were engaged tended to not be afraid of vaccine side effects. Lack of confidence connected to the risk of side effects of the vaccine is one of the most frequently noted as a barrier to vaccination and present among HCWs in different countries [32]. On the other hand, having a higher education or a partner seems to be a protecting factor. Another argument for vaccination avoidance is that influenza is not a dangerous disease. This belief tended to be more present among those with a longer length of service, probably because of relying on their personal long work experience. Unfortunately, this is a widespread thought in different health settings in different countries and much more must be done to invert this perception [33,34,35,36,37,38,39]. Furthermore, some of the HCWs, in particular those aged 40 years or older, tended to believe that asymptomatic cannot spread flu. It is calculated that one in three individuals with influenza is asymptomatic [40], and according to mathematical transmission models, the proportion of transmission of these individuals is one-third to one-half that of symptomatics [41]. So, this aspect must also be considered in order to increment the awareness of the HCWs towards the importance of vaccination and the risk of spreading influenza among patients.

When asked if mandatory vaccination for HCWs would be a fear rule, two-thirds of the respondents agreed with it. Particularly, 88.8% would undergo mandatory vaccination if it is offered free of charge and in the workplace. Furthermore, physicians were more compliant than other health professionals (94.8% vs. 82.4%) in line with other studies [25]. A recent systematic review and meta-analyses by Gualano et al. found a positive association between vaccine acceptance and mandatory flu vaccination. The high percentage of those favorable with compulsory vaccination could be explained by the fact that a high proportion of HCWs in our study support influenza vaccination. In their study, Gualano et al. reported that 61% of the pooled HCWs accepted mandatory policy with differences depending by continent (higher rates in Asia and lower in Europe) and health professional category (lower among nurses than other categories) [40].

Despite this degree of adherence, 40.7% of the healthcare providers consider mandatory vaccination as unethical because it violates the individual’s choice. However, at the same time, health providers have professional obligations to protect patients, especially vulnerable people who are a large proportion of those recovered in hospitals, and this aspect should overcome personal freedom. Durando et al. found that one of the major drivers for getting flu vaccination annually was the fact that it is considered an ethical duty by HCWs. This aspect needs to be emphasized in order to increase adherence to flu vaccination [41].

One quarter of the participants, especially physicians or HCWs with a longer length of service, are against mandatory vaccination if they work in low-intensity wards or if they are healthy, while engaged people think that flu vaccination should only be recommended. Other studies presented similar results [33,37].

Even though in this case there might be a reduced risk for patients, other aspects should be considered such as workday lost or the risk that COVID-19 symptoms are mistaken with flu and vice versa, with consequences for the patient, the caregiver, and the healthcare facility.

Another finding in our study is that the great majority of physicians (93.9%) see flu vaccination as a professional responsibility considering the risk of transmitting influenza to patients; 82.9% of nurses share the same opinion; while HCWs with a longer length of service tend to not agree. As has been shown by several studies, physicians tend to be more compliant about receiving flu vaccination. For doctors, it is important to promote vaccination because they consider themselves as responsible for the citizen’s health education. This finding takes on greater importance considering that according to the essay of the National Institute of Health in Italy, only 50% of the elderly are vaccinated against influenza and less than 10–20% of people belong to other high-risk groups, including children with diseases.

Moreover, especially those aged 40 years or older, think that health education programs could be useful for getting information about flu vaccination. This can serve as a cue to plan, organize, and implement educational programs to inform and train HCWs about influenza vaccination with the aim of bringing down barriers and increase vaccination coverage.

### 4.1. Implications for Practice, Research and Policy

From a practical point of view, some considerations are needed. Almost 90% of participants believe that health education programs organized in the workplace may be useful to learn about Influenza vaccination, and flu vaccination should be mandatory if it were offered directly at the workplace. This answer confirms how multiple actions in education, promotion, and access to vaccination, can be useful to increase willingness to be vaccinated and coverage rates [27], with the coordinated effort of the hospital management, occupational medicine, and vaccination units. Continuing Medical Education (CMO) programs on vaccinations must be delivered continuously, since there is evidence that the combination of an educational and a promotional element is the most effective tool in increasing the influenza vaccination coverage among HCWs, with the effect of doubling the vaccination coverage for each season [42], providing specific information for this category of workers [43]. Possible new “Fluad-case”, intended as a generalized panic capable of compromising immunization campaigns and negatively affecting disease-related outcomes, must be avoided especially among HCWs, in which there is high probability of generating extremely serious health and economic losses for individuals and society [44].

The fear for flu vaccine side effects and for injection is still high. A recent meta-analysis was performed on the Safety and Efficacy of Spray Intranasal Live Attenuated Influenza Vaccine [45]. This systematic review, based on 488 participants coming from 22 studies, demonstrated a higher probability of seroconversion compared with placebo and considering the A/H1N1 serotype in healthy adults (OR = 2.26; 95% CI = 1.12–4.54). Considering the side effects, none of the analyzed symptoms showed a higher risk of events compared to subjects who received placebo, other than local symptoms, i.e., sore throat, nasal congestion, and rhinorrhea. These results are promising and must be further developed in order to face vaccine hesitancy.

From the policy point of view, some considerations arise. Three quarter of the participants believe it is fair that influenza vaccination is mandatory for healthcare professionals, even if 40% answered that forcing healthcare professionals to get vaccinated is unethical as it violates the individual’s choice. If we consider the principles of bioethics in which welfare concerns outweigh concerns about autonomy, and by examining the virtues of the healing professions and the derivative institutional obligations, we agree with Tilburt et al. who argue that healthcare institutions are due to achieve adequate vaccination rates, and if needed, mandatory vaccination [46]. In Italy, however, until now, the flu vaccination has not been mandatory for HCWs.

Some final thoughts are needed. Even if not mandatory, the flu vaccination campaign 2020–2021 reached a vaccination rate of 63% among HCWs of the teaching hospital and this result is clearly related to SARS-CoV-2 pandemic that heavily affected the potential impact on the “background” acceptance of vaccination (2–4% vaccination rates in years just before the pandemic).

### 4.2. Limitations and Strengths

Our study has some limitation. The main one is its cross-sectional nature, which allows us to describe general associations but not to determine the cause-and-effect relationship between the predictor variables and the dependent variable. Second, our study was limited to one hospital in one city and started in a period in which the anti-SARS-CoV-2 vaccine was not available yet, so our results cannot be generalized to the region or country as whole. Furthermore, the presence of response bias due to the phrasing of some questions investigating knowledge, attitudes, and behavior of HCWs, as they could have led the answer of the participants. Moreover, since 25.1% of the potential participants entered the study, selection bias cannot be excluded, even if the proportion of job activities were similar to that of the total number of employees. Finally, some variables such as COVID-19 care provision, past history of infections, and other clinical issues are not included in the study.

Nevertheless, some strengths should be mentioned such as high rate of response, number of the participants in the survey, and the number of items investigated.

## 5. Conclusions

The findings in our study allow us to understand the variants of adherence to seasonal flu vaccination among Italian HCWs. These results could also be used to improve any future promotion campaigns to overcome the identified barriers to immunization. Immunizing health professionals means, on the one hand, protecting the health of patients and operators, and on the other hand, limiting the spread of diseases, especially during epidemics. Although vaccination is active, a small percentage of health workers are still opposed to vaccination. In this case, it is important to support the promotion of influenza vaccination through communication and health education programs organized in hospitals to obtain information on vaccination that would increase vaccination coverage. The COVID-19 pandemic serves as a good reminder for healthcare professionals to get the flu vaccination.

## Figures and Tables

**Table 1 vaccines-10-01341-t001:** Sample descriptive date.

Variable	N (%) or Median (DS)
*Gender*	
Female	550 (63.1%)
Male	322 (36.9%)
*Age*	41.7 (14.0)
*Length of service*	16.9 (13.3%)
*Lenght of service in the organization they work for*	12.8 (12.4%)
*Civil status*	
Cohabitant	192 (22%)
Single	287 (32.9%)
Separated or divorced	58 (6.7%)
Married	335 (38.4%)
*Role*	
Physician	446 (51.1%)
Other healthcare worker	126 (14.4%)
Auxiliary	14 (1.6%)
Biologist	23 (2.6%)
Nurse	232 (26.6%)
Technician	31 (3.6%)

**Table 2 vaccines-10-01341-t002:** (**a**) Univariate analysis—what do HCWs think about influenza? Absolute values and percentages of the Yes answers are reported. (**b**) Univariate analysis—vaccine obligation. Absolute values and percentages of the Yes answers are reported. (**c**) Univariate analysis—promotion and vaccine hesitancy. Absolute values and percentages of the Yes answers are reported.

(**a**)												
**Variable**		*Influenza is a dangerous disease only for elderly people (age ≥ 65 years) and for those with previous illnesses*		*p*	*If I and/or my colleagues get sick with flu, difficulties may arise in reorganizing the staff shifts.*	*p*	*I can transmit the Influenza even if I am asymptomatic.*			*p*	*Flu is NOT a dangerous disease for healthy people.*	*p*
**Total**	No	278 (31.9)			53 (6.1)		153 (17.5)				333 (38.2)	
	Yes	549 (68.1)			819 (93.9)		719 (82.5)				539 (61.8)	
**Age**	<40	249 (58.7)		<0.001	398 (93.9)	0.979	382 (90.1)			<0.001	238 (56.1)	0.001
	>40	329 (77)			401 (93.9)		321 (75.2)				285 (66.7)	
**Gender**	F	368 (66.9)		0.316	522 (94.9)	0.111	443 (80.5)			0.053	342 (62.2)	0.769
	M	226 (70.2)			297 (92.2)		276 (85.7)				197 (61.2)	
**Civil status**	Not married	228 (66.1)		0.297	327 (94.8)	0.389	291 (84.3)			0.234	208 (60.3)	0.454
	Married	366 (69.4)			492 (93.4)		428 (81.2)				331 (62.8)	
**Job title**	Physicians	305 (68.4)		0.863	416 (93.3)	0.412	381 (85.4)			0.018	283 (63.5)	0.307
	Other Health Care Workers	289 (67.8)			403 (94.6)		338 (79.3)				256 (60.1)	
(**b**)												
**Variable**		*Influenza vaccination is a professional responsibility for healthcare professionals, who can inadvertently transmit the flu, endangering patients lives.*		*p*	*I would voluntarily undergo flu vaccination to protect myself and the people I come in contact with.*	*p*	*I would have mandatory flu vaccination if it were offered directly at the workplace.*			*p*	*It’s fair that influenza vaccination is mandatory for healthcare professionals.*	*p*
**Total**	No	100 (11.5)			91 (10.4)		98 (11.2)				225 (25.8)	
	Yes	772 (88.5)			781 (89.6)		774 (88.8)				647 (74.2)	
**Age**	≤40	386 (91)		**0.020**	383 (90.3)	0.595	389 (91.7)			**0.017**	321 (75.7)	0.338
	>40	367 (85.9)			381 (89.2)		370 (86.7)				311 (72.8)	
**Gender**	F	483 (87.8)		0.387	487 (88.5)	0.198	477 (86.7)			0.013	388 (70.5)	0.001
	M	289 (89.8)			294 (91.3)		297 (92.2)				259 (80.4)	
**Civil status**	Not married	314 (91)		0.063	315 (91.3)	0.174	312 (90.4)			0.206	260 (75.4)	0.525
	Married	458 (86.9)			466 (88.4)		462 (87.7)				387 (73.4)	
**Job title**	Physicians	419 (93.9)		**<0.001**	423 (94.8)	<0.001	423 (94.8)			<0.001	378 (84.8)	<0.001
	Other Health Care Workers	353 (82.9)			358 (84)		351 (82.4)				269 (63.1)	
**Variable**		*Forcing healthcare professionals to get vaccinated is unethical as it violates the individual’s choice.*		*p*	*Only elderly people (age ≥ 65 years) and those with previous illnesses should be required to undergo Influenza vaccination*	*p*	*Influenza vaccination should only be recommended for healthcare professionals*			*p*	*I do not have to be forced to undergo Influenza vaccination if I work in a low-risk Operating Unit.*	*p*
**Total**	No	517 (59.3)			582 (66.7)		588 (67.4)				640 (73.4)	
	Yes	355 (40.7)			290 (33.3)		284 (32.6)				232 (26.6)	
**Age**	≤40	158 (37.3)		0.052	115 (27.1)	<0.001	133 (31.4)			0.463	110 (25.9)	0.578
	>40	187 (43.8)			169 (39.6)		144 (33.7)				118 (27.6)	
**Gender**	F	238 (43.3)		0.044	182 (33.1)	0.892	182 (33.1)			0.667	163 (29.6)	0.008
	M	117 (36.3)			108 (33.5)		102 (31.7)				69 (21.4)	
**Civil status**	Not married	133 (38.6)		0.293	107 (31)	0.255	93 (27)			0.004	92 (26.7)	0.974
	Married	222 (42.1)			183 (34.7)		191 (36.2)				140 (26.6)	
**Job title**	Physicians	128 (28.7)		<0.001	143 (32.1)	0.444	138 (30.9)			0.294	84 (18.8)	<0.001
	Other Health Care Workers	227 (53.3)			147 (34.5)		146 (34.3)				148 (34.7)	
(**c**)												
**Variable**		*I don’t have to undergo Influenza vaccination if I’m healty.*	*p*	*Healthcare professionals must promote vaccination because they are responsible for the citizen’s health education*	*p*	*I’m afraid of the Influenza vaccination’s side effects*.	*p*	*I’m afraid of injection*	*p*	*Health education programs organized in the structure where I work may be useful for me to learn about Influenza vaccination.*		*p*
**Total**	No	649 (74.4)		81 (9.3)		695 (79.7)		816 (93.6)		136 (15.6)		
	Yes	223 (25.6)		791 (90.7)		177 (20.3)		56(6.4)		736 (84.4)		
**Age**	≤40	98 (23.1)	0.112	396 (93.4)	**0.010**	64 (15.1)	<0.001	28(6.6)	0.758	348 (82.1)		0.101
	>40	119 (27.9)		377 (88.3)		107 (25.1)		26(6.1)		368 (86.2)		
**Gender**	F	151 (27.5)	0.096	490 (89.1)	**0.031**	129 (23.5)	0.002	39 (7.1)	0.292	459 (83.5)		0.313
	M	72 (22.4)		301 (93.5)		48 (14.9)		17 (5.3)		277 (86)		
**Civil status**	Not married	77 (22.3)	0.075	322 (93.3)	**0.031**	67 (19.4)	0.602	31(9)	0.012	294 (85.2)		0.592
	Married	146 (27.7)		469 (89)		110 (20.9)		25 (4.7)		442 (83.9)		
**Job title**	Physicians	80 (17.9)	<0.001	431 (96.6)	**<0.001**	39 (8.7)	<0.001	23(5.2)	0.119	375 (84.1)		0.788
	Other Health Care Workers	143 (33.6)		360 (84.5)		138 (32.4)		33 (7.7)		361 (84.7)		
**Variable**		*It’s not necessary to undergo Influenza vaccination if you respect the hygiene rules (for example, washing your hands often, especially after sneezing/coughing or going to public places).*	*p*	*If I am vaccinated, I don’t risk transmitting flu to patients I take care of and to the people I come into contact with.*	*p*	*The Influenza vaccine can have dangerous side effects.*	*p*	*Influenza vaccination is important to prevent the disease.*	*p*	*Have you ever had the flu shot in your life?*		*p*
**Total**	No	730 (83.7)		211 (24.2)		664 (76.1)		49 (5.6)		320 (36.7)		
	Yes	142 (16.3)		661 (75.8)		208 (23.9)		823 (94.4)		552 (63.3)		
**Age**	≤40	71 (16.7)	0.677	309 (72.9)	0.070	93(21.9)	0.249	403 (95)	0.386	244 (57.5)		<0.001
	>40	67 (15.7)		334 (78.2)		108 (25.3)		400 (93.7)		293 (68.6)		
**Gender**	F	98 (17.8)	0.109	422 (76.7)	0.405	135 (24.5)	0.531	512 (93.1)	0.031	342 (62.2)		0.369
	M	44 (13.7)		239 (74.2)		73 (22.7)		311 (96.6)		210 (65.2)		
**Civil status**	Not married	64 (18.6)	0.143	256 (74.2)	0.372	79 (22.9)	0.593	328 (95.1)	0.473	212 (61.4)		0.358
	Married	78 (14.8)		405 (76.9)		129 (24.5)		495 (93.9)		340 (64.5)		
**Job title**	Physicians	40(9)	<0.001	374 (83.9)	<0.001	64 (14.3)	<0.001	434 (97.3)	<0.001	331 (74.2)		<0.001
	Other Health Care Workers	102 (23.9)		287 (67.4)		144 (33.8)		389 (91.3)		221 (51.9)		

**Table 3 vaccines-10-01341-t003:** (**a**) Multivariate analysis—what do HCWs think about influenza? (**b**) Multivariate analysis—vaccine obligation. (**c**) Multivariate analysis—promotion and vaccine hesitancy.

(**a**)					
**Variable**	*Influenza is a dangerous disease only for elderly people (age ≥ 65 years) and for those with previous illnesses*		*Flu is NOT a dangerous disease for healthy people.*	*I can transmit the Influenza even if I am asymptomatic.*	*If I and/or my colleagues get sick with flu, difficulties may arise in reorganizing the staff shifts.*
	OR (95% CI)		OR (95% CI)	OR (95% CI)	OR (95% CI)
**Gender Females (ref. Males)**	-		-	0.67 (0.45–1.00)	1.57 (0.90–2.75)
**Age over 40 (ref. under or equal to 40)**	1.69 (1.07–2.67)		-	0.46 (0.26–0.81)	-
**Married/Living with a partner (ref. Single, Divorced and Widowed)**	-		-	-	-
**Doctor (ref. other Health Care Workers)**	-		-	1.42 (0.98–2.05)	-
**Lenght of service in the organization they work for**	1.56 (0.97–2.50)		1.65 (1.24–2.18)	0.63 (0.36–1.08)	-
(**b**)					
**Variable**	*Influenza vaccination is a professional responsibility for healthcare professionals, who can inadvertently transmit the flu, endangering patients lives.*		*I would voluntarily undergo flu vaccination to protect myself and the people I come in contact with.*	*I would have mandatory flu vaccination if it were offered directly at the workplace.*	*It’s fair that influenza vaccination is mandatory for healthcare professionals.*
	OR (95% CI)		OR (95% CI)	OR (95% CI)	OR (95% CI)
**Gender Females (ref. Males)**	-		-	0.65 (0.40–1.08)	0.71 (0.50–1.01)
**Age over 40 (ref. ≤ 40)**	-		-	-	-
**Married/Living with a partner (ref. Single, Divorced and Widowed)**	-		0.67 (0.42–1.06)	-	-
**Doctor (ref. other Health Care Workers)**	3.21 (2.01- 5.13)		3.56 (2.17–5.85)	3.61 (2.19–5.97)	3.03 (2.18–4.22)
**Lenght of service in the organization they work for**	0.45 (0.29–0.70)		-	0.44 (0.28–0.68)	-
**Variable**	*Forcing healthcare professionals to get vaccinated is unethical as it violates the individual’s choice.*		*Only elderly people (age ≥ 65 years) and those with previous illnesses should be required to undergo Influenza vaccination*	*Influenza vaccination should only be recommended for healthcare professionals*	*I do not have to be forced to undergo Influenza vaccination if I work in a low-risk Operating Unit.*
	OR (95% CI)		OR (95% CI)	OR (95% CI)	OR (95% CI)
**Gender Females (ref. Males)**	-		-	-	-
**Age over 40 (ref. ≤ 40)**	-		-	-	-
**Married/Living with a partner (ref. Single, Divorced and Widowed)**	-		-	1.54 (1.15–2.08)	-
**Doctor (ref. other Health Care Workers)**	0.35 (0.27–0.47)		-	-	0.43 (0.32–0.59)
**Lenght of service in the organization they work for**	1.40 (1.06–1.86)		1.78 (1.34–2.36)	-	-
(**c**)					
**Variable**	*Influenza vaccination is important to prevent the disease.*		*It’s not necessary to undergo Influenza vaccination if you respect the hygiene rules (for example, washing your hands often, especially after sneezing/coughing or going to public places).*	*If I am vaccinated, I don’t risk transmitting flu to patients I take care of and to the people I come into contact with.*	*I’m afraid of the Influenza vaccination’s side effects*.
	OR (95% CI)		OR (95% CI)	OR (95% CI)	OR (95% CI)
**Gender Females (ref. Males)**	0.54 (0.26–1.11)		-	1.45 (1.04–2.04)	-
**Age over 40 (ref. ≤ 40)**	-		-	1.47 (1.07–2.03)	1.90 (1.33–2.71)
**Married/Living with a partner (ref. Single, Divorced and Widowed)**	-		-	-	-
**Doctor (ref. other Health Care Workers)**	3.00 (1.52–5.91)		0.32 (0.21–0.47)	2.73 (1.96–3.82)	0.20 (0.13–0.29)
**Lenght of service in the organization they work for**	-		-	-	-
**Variable**	*I’m afraid of injection*	*The Influenza vaccine can have dangerous side effects.*	*Healthcare professionals must promote vaccination because they are responsible for the citizen’s health education*	*Health education programs organized in the structure where I work may be useful for me to learn about Influenza vaccination.*	*Have you ever had a flu shot in your life?*
	OR (95% CI)	OR (95% CI)	OR (95% CI)	OR (95% CI)	OR (95% CI)
**Gender Females (ref. Males)**	-	-	-	-	-
**Age over 40 (ref. ≤ 40)**	-	-	-	2.15 (1.14–4.08)	1.69 (1.27–2.25)
**Married/Living with a partner (ref. Single, Divorced and Widowed)**	0.50 (0.29–0.87)	-	0.58 (0.34–1.00)	0.82 (0.55–1.22)	
**Doctor (ref. other Health Care Workers)**	-	0.33 (0.24–0.46)	5.49 (3.06–9.84)	-	2.72 (2.04–3.63)
**Lenght of service in the organization they work for**	-	-	0.60 (0.37- 0.99)	0.62 (0.33–1.18)	-

## Data Availability

Data are available upon request.

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
