# Peer review of "Flu Vaccination among Healthcare Professionals in Times of COVID-19: Knowledge, Attitudes, and Behavior"

_vaccines, 2022, doi:10.3390/vaccines10081341_

Round 1

Reviewer 1 Report

Thank you for your submission. Some comments-

1. There is nothing new about this study and there are hundreds of such papers available online now. Also, study limitations and implications for practice, research, or policy are missing or sparsely described

2. The language is seriously flawed (e.g. COVID is used instead of COVID-19 and there are tons of such errors).

3. The statistics are mostly incorrectly done (e.g. regression analysis- what is the reference group is not clear, in univariate analysis - p values are given for yes. vs no on first question- what was the comparison group?

4. Important variables such as COVID-19 care provision, past history of infections, etc are not included in the study.

5. The study measures are neither reliable not valid, so all findings stand on weak grounds.

6. How was the sample size determined? what power analysis was conducted? how were type 1 and 2 errors controlled for? These is no such detail

Author Response

Reviewer 1

  1. There is nothing new about this study and there are hundreds of such papers available online now. Also, study limitations and implications for practice, research, or policy are missing or sparsely described

Answer: the reviewer is right, there are several papers on this issue. However, the studies conducted and published in Italy were only three and report low adherence to flu vaccination even if improved compared to pre-pandemic period.

We included the following statements in the Discussion section:

Italy is an interesting country to consider on this issue, due to the low attitudes and adherence toward flu vaccination (19). The flu coverage rate in the pre-pandemic period was between 4 and 13% (20), and this survey demonstrate the high degree of flu vaccine trust and willingness to get vaccinated, much more than those reported in other studies (between 40.8% and 53.8%). (21-23).”

Moreover, we added a paragraph concerning Implications for practice, research and policy, as follows:

“From a practical point of view, some considerations are needed. Almost 90% of participants believe that Health education programs organized in the workplace may be useful to learn about Influenza vaccination and flu vaccination should be mandatory if it were offered directly at the workplace. This answer confirms how multiple actions in Education, Promotion, and Access to vaccination, can be useful to increase willingness to be vaccinated and coverage rates (20), with the coordinated effort of the Hospital Management, Occupational Medicine and Vaccination Units. Continuing Medical Education programs on vaccinations must be delivered continuously, since there is evidence that the combination of an educational and a promotional element is the most effective tool in increasing the influenza vaccination coverage among HCWs, with the effect of doubling the vaccination coverage for each season (37).

The fear for flu vaccine side effects and for injection is still high.  A recent meta-analysis was performed on Safety and Efficacy of Spray Intranasal Live Attenuated Influenza Vaccine (38). This systematic review, based on 488 participants coming from 22 studies demonstrated a higher probability of seroconversion compared with placebo and considering the A/H1N1 serotype in healthy adults (OR = 2.26; 95% CI = 1.12–4.54). Considering the side effects, none of the analysed symptoms showed a higher risk of events compared to subjects who received placebo, other than local symptoms, i.e., sore throat, nasal congestion, and rhinorrhea. These results are promising and must be further developed in order to face with vaccine hesitancy.

From the policy point of view, some considerations arise. Three quarter of the participants believe it’s fair that influenza vaccination is mandatory for healthcare professionals, even if 40% answered that Forcing healthcare professionals to get vaccinated is unethical as it violates the individual's choice. If we consider the principles of bioethics in which welfare concerns outweigh concerns about autonomy, and by examining the virtues of the healing professions and the derivative institutional obligations we agree with Tilburt et al that argue healthcare institutions are due to achieve adequate vaccination rates, and if needed, mandatory vaccination (39). In Italy, however, until now the flu vaccination is not mandatory for HCWs.”

  1. The language is seriously flawed (e.g. COVID is used instead of COVID-19 and there are tons of such errors).

Answer: we replace Covid with Covid-19 in the title and in the abstract.

We corrected the language as requested.

  1. The statistics are mostly incorrectly done (e.g. regression analysis- what is the reference group is not clear, in univariate analysis - p values are given for yes. vs no on first question- what was the comparison group?

Answer: the reviewer is right. We mistakenly reported p-values in the first column of the univariate analysis.

Concerning the multivariate analysis, for increase the clarity of the manuscript, the reference groups are now reported in the table.

  1. Important variables such as COVID-19 care provision, past history of infections, etc are not included in the study.

Answer: we agree with the reviewer. We included these issues as potential limitations of the study. The following sentence was added:

“Finally, some variables such as COVID-19 care provision, past history of infections, and other clinical issues are not included in the study.”

  1. The study measures are neither reliable not valid, so all findings stand on weak grounds.

Answer: the reviewer is right in the sense that the statistics of the questionnaire were not mentioned in the previous version of the manuscript. We included in this revised version the statistics on reliability on the questionnaire.

The following statement was added:

“The questionnaire showed a good reliability (Cronbach alpha = 0.738)”

  1. How was the sample size determined? what power analysis was conducted? how were type 1 and 2 errors controlled for? These is no such detail

Answer: many thanks for this comment. We included in the revised version the sample size calculations. The following statement was added in the Methods section:

“Sample size calculation was carried out using EpiCalc2000 with the following parameters:

Proportion of willingness to get vaccinated among HCWs: 47,00% (mean between values reported in other studies)[x, z]   

Null hypothesis value: 52,00%    

Significance: 0,05     

Power: 80%       

The Sample size needed was 769. In order to increase the power, a 10% increase was added. So, the final sample size calculation gave the need to recruit at least 846 HCWs.”        

You can also see the attachment

Reviewer 2 Report

Estimated Dr. COLAPRICO,

I've read your report with great interest. Albeit not particularly innovative (as  stated by the study Authors in both introduction and discussion) this study is well interesting for all potential readers, not only Italian ones but also international readers, because of the following pros:

a) it was performed during the COVID-19 pandemic but shortly before the availability of SARS-CoV-2 vaccine;

b) it encompasses a relatively large study population but from a single centre, eventually minimizing potential heterogeneities of the targeted professionals.

Despite these strengths, some shortcomings affect the overall acceptance of this study. I think that the Authors could fix all the following issues from the present Reviewer quite easily, as not requiring new or improved analysis of data but only formal issues.

More precisely:

1) how was the questionnaire delivered? were all the HCWs targeted and potentially recruited? what about the participation rate calculated over the total number of employees from the parent Hospital?

2) some typos affect the main text: citation of Dini et al (16) does not contain the estimate reported by the authors, please double check; row 78 lacks of a ")"; comma based notation of decimals is reported instead of point based one; there is inconsistent reporting of decimals (sometimes 1, sometimes 2: please make a choice and stick with it)

3) the acronym HCW should be solved in the captions of tables at least the first time it is reported

4) as 5 "a-priori" variables were identified to be explored as explanatory ones (i.e. gender, age, marriage, qualification, seniority) Authors should preventively report the rationale for this choice either in the introduction or in the materials and methods section; reader familiar with Vaccine Hesitancy may easily understand why Authors have stressed these factors, but those more unfamiliar may wonder why gender and seniority, for example, may influence the acceptance of a vaccine. In this regard, some commentaries on case studies such as "Fluad" and the episodic imbalance between seasonal influenza formulates and actually circulating strains (with resulting scarce confidence from more "seasoned" professionals) may improve the overall quality of the paper.

5) I have some doubts about including all HCWs as a whole, as in fact your sample encompasses physicians and alleged medical personnel whose expertise and competence on vaccines and their reliability may be quite heterogenous. This potential issue should be acknowledged as a limit in the discussion section. Similarly, the convenience sampling and its potential consequences should be more extensively discussed.

6) Authors have not discussed the potential role of SARS-CoV-2 pandemic both in the potential generalizability of the results and on the potential impact of the pandemic on the "background" acceptance of vaccination. In fact, some insights on the vaccination campaign 2020-2021 compared to the previous ones would similarly improve the interest of this study allowing a better comparison with similar studies from other European Countries.

Author Response

Reviewer 2

Estimated Dr. COLAPRICO,

I've read your report with great interest. Albeit not particularly innovative (as  stated by the study Authors in both introduction and discussion) this study is well interesting for all potential readers, not only Italian ones but also international readers, because of the following pros:

  1. a) it was performed during the COVID-19 pandemic but shortly before the availability of SARS-CoV-2 vaccine;
  2. b) it encompasses a relatively large study population but from a single centre, eventually minimizing potential heterogeneities of the targeted professionals.

Answer: many thanks for these considerations.

Despite these strengths, some shortcomings affect the overall acceptance of this study. I think that the Authors could fix all the following issues from the present Reviewer quite easily, as not requiring new or improved analysis of data but only formal issues.

More precisely:

  • how was the questionnaire delivered? were all the HCWs targeted and potentially recruited? what about the participation rate calculated over the total number of employees from the parent Hospital?

Answer: many thanks for these comments. We specified in the methods that the questionnaire was delivered online inviting the potential participants via email.

In the Limitations of the study we reported

“Moreover, since 25.1% of the potential participants entered the study, selection bias cannot be excluded, even if the proportion of job activities were similar to that of the total number of employees”.

2) some typos affect the main text: citation of Dini et al (16) does not contain the estimate reported by the authors, please double check; row 78 lacks of a ")"; comma based notation of decimals is reported instead of point based one; there is inconsistent reporting of decimals (sometimes 1, sometimes 2: please make a choice and stick with it)

Answer: we checked for these typos as suggested and made amendments.

3) the acronym HCW should be solved in the captions of tables at least the first time it is reported

Answer: done as suggested

4) as 5 "a-priori" variables were identified to be explored as explanatory ones (i.e. gender, age, marriage, qualification, seniority) Authors should preventively report the rationale for this choice either in the introduction or in the materials and methods section; reader familiar with Vaccine Hesitancy may easily understand why Authors have stressed these factors, but those more unfamiliar may wonder why gender and seniority, for example, may influence the acceptance of a vaccine. In this regard, some commentaries on case studies such as "Fluad" and the episodic imbalance between seasonal influenza formulates and actually circulating strains (with resulting scarce confidence from more "seasoned" professionals) may improve the overall quality of the paper.

Answer: we really thank the reviewer for these suggestions.

We made some specification concerning the introduction of gender and seniority as explanatory variables. The following sentences were added.

“The choice of gender as possible explanatory variable is due to the highest likelihood of vaccine hesitancy among women (21), while age and job experience were chosen since increasing levels of these variables are associated to different odds of getting flu vaccination (22-24).”

The need for Continuing Medical education is now better specified in the Discussion section.

Moreover, we added a commentary on the Fluad case that is interesting for the Discussion.

“Possible new “Fluad-case”, intended as a generalized panic capable of compromising immunization campaigns and negatively affecting disease related outcomes, must be avoided especially among HCWs, in which there is high probability of generating extremely serious health and economic losses for individuals and society.”

5) I have some doubts about including all HCWs as a whole, as in fact your sample encompasses physicians and alleged medical personnel whose expertise and competence on vaccines and their reliability may be quite heterogenous. This potential issue should be acknowledged as a limit in the discussion section. Similarly, the convenience sampling and its potential consequences should be more extensively discussed.

Answer: we included these issues in the Discussion section.

6) Authors have not discussed the potential role of SARS-CoV-2 pandemic both in the potential generalizability of the results and on the potential impact of the pandemic on the "background" acceptance of vaccination. In fact, some insights on the vaccination campaign 2020-2021 compared to the previous ones would similarly improve the interest of this study allowing a better comparison with similar studies from other European Countries.

Answer: we agree with the reviewer. We included in the Discussion section some insight on the flu vaccination campaign 2020-2021 compared to the previous ones.

The following sentences were added:

“Some final thoughts are needed. Even not mandatory, the flu vaccination campaign 2020-2021 reached a vaccination rate of 63% among HCWs of the teaching hospital and this result is clearly related to SARS-CoV-2 pandemic that heavily affected the potential impact on the "background" acceptance of vaccination (2-4% vaccination rates in years just before the pandemic). ”

“Second, our study was limited to one hospital in one city and started in a period in which the anti SARS-CoV-2 vaccine was not available yet, so our results cannot be generalized to the region or country as whole”

You can also see the attachment.

Round 2

Reviewer 1 Report

Thanks for the revisions.

Author Response

Many thanks for having accepted the new version of the manuscript